# A Macro-Level Association of Vaccination Rate with the Number of Confirmed COVID-19 Cases in the United States and Japan

**DOI:** 10.3390/ijerph19127435

**Published:** 2022-06-17

**Authors:** Hiroyuki Noda

**Affiliations:** 1Public Health, Department of Social Medicine, Graduate School of Medicine, Osaka University, Osaka 565-0871, Japan; hiroyuki-noda@umin.net; 2Cabinet Secretariat, Tokyo 100-8968, Japan

**Keywords:** COVID-19, vaccination, Unites States, Japan

## Abstract

Aiming to evaluate a macro-level association of vaccination rate as well as booster vaccination rate with the number of confirmed COVID-19 cases in the United States and Japan, a cross-sectional study was conducted using data in each jurisdiction. Data on the total number of people who were fully vaccinated as of the end of December 2021, data on the total number of people who have received a booster dose as of the end of March 2022 and data on the cumulative number of confirmed COVID-19 cases were obtained from the website of the national governments. A generalized regression model was used to examine the association. This study showed that a higher vaccination rate was associated with a lower number of confirmed COVID-19 cases per year in 2021 for both the United States and Japan. The number of confirmed COVID-19 cases per 1000 population per year (95% confidence intervals) as a 1% increment of the vaccination rate was −0.74 (−1.29, −0.20), *p* = 0.007 for the United States and −1.48 (−1.95, −1.00), *p* < 0.0001 for Japan. A similar association was observed for the booster vaccination rate in 2022, although the association was attenuated in a multivariable model, particularly for the United States. This study provided macro-level evidence that vaccination may reduce the number of confirmed COVID-19 cases.

## 1. Introduction

From the historical point of view, vaccines have been essential for the eradication or drastic reduction in the incidence of common diseases and developed against a wide range of pathogens, including SARS-CoV-2 [1], but many vaccines confer only short-term and modest levels of individual protection [2,3,4]. Therefore, a vaccine is not always the ultimate solution; vaccination has been implemented as one of the countermeasures against some infectious diseases to provide imperfect but widespread protection to the masses. Even in the pandemic of Coronavirus disease 2019 (COVID-19), vaccination has been implemented as one of the crucial countermeasures against SARS-CoV-2 and COVID-19.

In the COVID-19 pandemic, some previous studies indicate that vaccination may reduce the risk of SARS-CoV-2 infections in individual-level analyses [5,6,7,8,9,10,11,12,13,14,15,16,17,18,19,20,21,22]. Although the waning of vaccine protection and the immune escape of variants, such as Delta (B.1.617.2) and Omicron variants (B.1.1.529), has been particularly worrying [23], these studies indicate that vaccination has at least modest levels of individual protection. In fact, a previous study shows that the effectiveness against the Omicron variant was 71.6% at 14–60 days and 47.4% at >60 days after vaccination [20], and another study shows that a booster dose of the mRNA vaccine (i.e., BNT162b2 and mRNA-1273) was effective in preventing symptomatic infections by the Delta variant, but less effective against the Omicron variant, although the effectiveness against the Omicron variant was 49.4% for BNT162b2 mRNA vaccine and 47.3% for mRNA-1273 mRNA vaccine [21]. However, there is limited evidence for the masses that show that vaccination reduced the number of confirmed COVID-19 cases in a macro-level analysis (i.e., an analysis using population, but not individual, as each observation in the dataset), and this inconsistency may affect vaccine acceptance. Moreover, mixed messages about the effectiveness of the vaccine (e.g., the difference in effectiveness between SARS-CoV-2 infections and COVID-19 severe diseases) may affect the vaccine acceptance because the effectiveness of the vaccination at reducing SARS-CoV-2 infections may be weaker than that at reducing COVID-19 severe diseases [22]. In fact, a previous study showed that vaccine efficacy contributed to the intent to vaccinate [24], and it is claimed that Western countries with a higher vaccination rate have a higher number of confirmed COVID-19 cases to deny the effectiveness of vaccination against the COVID-19 pandemic. This kind of macro-level viewpoint is intuitively accepted by the public but is more likely to be affected by potential confounding factors such as population size and ideology in each country.

The United States and Japan are both democratic countries with similar size populations and Gross Domestic Product per capita, and a lot of national comparisons have been historically conducted on health topics [25,26]. These two countries have a similar number of political entities, which allows us to conduct a macro-level analysis in each country, leading to reduced bias as compared with that of a simple international comparison.

Therefore, this study conducted a macro-level analysis regarding an association of vaccination rate with the number of confirmed COVID-19 cases in the United States and Japan.

## 2. Materials and Methods

### 2.1. Study Materials

A cross-sectional study was conducted using data from each jurisdiction (i.e., 50 states and the District of Columbia for the United States and 47 prefectures for Japan) for the United States and Japan.

An association of vaccination rate with the number of confirmed COVID-19 cases was examined using data from each jurisdiction between January and December 2021 in the United States and Japan. The data on the total number of people who were fully vaccinated (i.e., the number of people who were vaccinated with the second dose of a two-dose vaccine or one dose of a single-dose vaccine) and data on the cumulative number of confirmed COVID-19 cases as of the end of December 2021 were obtained from the website of national governments [27,28]. Data on the total number of people who were fully vaccinated in each jurisdiction where the recipient lives were obtained from the website of the Centers for Disease Prevention and Control (CDC) [27] for the United States and the website of the Prime Minister’s Office [28] for Japan. Data on the cumulative number of confirmed COVID-19 cases were obtained from the website of CDC [29] for the United States and the website of the Ministry of Health, Labour and Welfare (MHLW) [30] for Japan. Data regarding vaccination published on December 27 were used for Japan because the data, including the number regarding health professionals, were not updated after December 28 in 2021. The population in each jurisdiction was used from the latest data on Population Estimates before the COVID-19 pandemic (i.e., 1 July 2019 for the United States and 1 October 2019 for Japan).

The association of the booster vaccination rate with the number of confirmed COVID-19 cases was examined using data from each jurisdiction between January and March 2022 in the United States and Japan. The data on the total number of people who were fully vaccinated as of the end of March 2022 were also used to adjust as a potential confounding factor. The data on the total number of people who were fully vaccinated, the data on the total number of people who have received a booster dose and the data on the cumulative number of confirmed COVID-19 cases as of the end of March 2022 were obtained from the websites of the national governments [27,28,29,30,31].

Aiming to show the global position of the United States and Japan, in addition to the difference between intranational and international associations of vaccination rate with the number of confirmed COVID-19 cases, a simple international comparison was also conducted to examine an association between vaccination rate and booster vaccination rate with the number of confirmed COVID-19 cases in the world. Data on the total number of people who were fully vaccinated, data on the total number of people who have received a booster dose and data on the cumulative number of confirmed COVID-19 cases in each country and area as of the end of December 2020, as of the end of December 2021 and as of the end of March 2022 were obtained from the website of Our World in Data [32]. A total of 228 countries and areas were identified in the dataset, and countries and areas without data on the total number of people who were fully vaccinated and those who have received a booster dose within two weeks (i.e., 14 days) before the last day of each year (i.e., December 31, 2021 and March 31, 2022) and the data on the cumulative number of confirmed COVID-19 cases as of the end of the last day of each year (i.e., December 2020, December 2021 and March 2022) were excluded from each analysis. Finally, 176 counties and areas were used for an international comparison of the vaccination rate in 2021, and 138 counties and areas were used for that of the booster vaccination rate in 2022.

Informed consent was not required in this study because publicly available data were used.

### 2.2. Statistical Analysis

Based on data from each jurisdiction (i.e., 50 states and the District of Columbia for the United States and 47 prefectures for Japan), a generalized regression model was used to examine the association of vaccination rate (i.e., the percentage of people who were fully vaccinated with the second dose of a two-dose vaccine or one dose of a single-dose vaccine) with the number of confirmed COVID-19 cases per 1000 population per year in 2021. In the calculation of the vaccination rate, the overall population was used as the denominator. The covariate was not included in the crude model, and both the vaccination rate and booster vaccination rate were included in the multivariable model as each covariate. An adjusted R-square was also calculated for each country. As sensitivity analysis, the log-linear model was also used in this analysis.

In an analysis of booster vaccination, the generalized regression model was similarly used to examine an association of booster vaccination rate (i.e., the percentage of people who have received a booster dose) with the number of confirmed COVID-19 cases per 1000 population per year in 2022, adjusted for vaccination rate (i.e., the percentage of people who were fully vaccinated with the second dose of a two-dose vaccine or one dose of a single-dose vaccine).

The variance inflation factor (VIF) of each variable was calculated to diagnose multicollinearity [33]. VIFs above 5 were considered as the presence of multicollinearity [34].

In a simple international comparison, the generalized regression model was used to examine an association of vaccination rate and booster vaccination rate with the number of confirmed COVID-19 cases per 1000 population per year. The vaccination rate as of the last day within 2 weeks before the end of December 2021 and the cumulative number of confirmed COVID-19 cases in 2021 was used for an international comparison of vaccination rate, and the booster vaccination rate as of the last day within 2 weeks before the end of March 2022 and the cumulative number of confirmed COVID-19 cases in 2022 was used for that of booster vaccination rate.

All statistical tests were 2-sided and conducted using SAS, version 9.4 (SAS Institute, Cary, NC, USA). *p*-values below 0.05 were considered statistically significant.

## 3. Results

### 3.1. A Simple International Comparison

In a simple international comparison using data from 176 countries and areas in 2021, a higher vaccination rate was associated with a higher number of confirmed COVID-19 cases per 1000 population per year in the world (Figure 1). The number of confirmed COVID-19 cases per 1000 population per year (95% confidence intervals), in a 1% increment of vaccination rate, was 1.05 (0.79, 1.32), *p* < 0.0001.

A similar international association was observed for the booster vaccination rate using data from 138 countries and areas in 2022 (Figure 2). The number of confirmed COVID-19 cases per 1000 population per year, in a 1% increment of the booster vaccination rate, was 10.10 (8.10, 12.10), *p* < 0.0001.

### 3.2. A Macro-Level Analysis in the United States and Japan

A total of 98 jurisdiction’s data were used for a macro-level analysis in the United States and Japan (Table 1). The number of people who were fully vaccinated, as well as the vaccination rate, mainly increased in 2021, and the vaccination rate was higher in Japan than that in the United States. On the other hand, the number of people who have received a booster dose as well as the booster vaccination rate was higher for the United States than that for Japan in 2021, and the booster vaccination rate for Japan surpassed that for the United States in 2022. The number of confirmed COVID-19 cases per year for the United States was higher than that for Japan throughout the study period.

Table 2 shows the association of the vaccination rate (i.e., the percentage of people who were fully vaccinated with the second dose of a two-dose vaccine or one dose of a single-dose vaccine) as of the end of December 2021 with the number of confirmed COVID-19 cases per 1000 population per year in 2021. In 2021, a higher vaccination rate was associated with a lower number of confirmed COVID-19 cases per year both in the United States and Japan, and the number of confirmed COVID-19 cases per 1000 population per year (95% confidence intervals) in a 1% increment of the vaccination rate was −0.74 (−1.29, −0.20), *p* = 0.007, for the United States and −1.48 (−1.95, −1.00), *p* < 0.0001, for Japan (*p* for interaction was 0.31). The adjusted R-square was 0.11 for the United States and 0.43 for Japan.

In the log-linear model, the associations were not altered substantially, although the interaction becomes large. The percent change in a 1% increment of the vaccination rate was −0.7% (−1.2%, −0.2%), *p* = 0.008, for the United States and −13.7% (−16.3%, −11.1%), *p* < 0.0001 for Japan (*p* for interaction was 0.003).

Figure 3 graphically demonstrates that the national difference in the number of confirmed COVID-19 cases between the United States and Japan was larger than the effect of the vaccination rate (Figure 3). The difference in mean values in the number of confirmed COVID-19 cases per 1000 population per year (95% confidence intervals) was 96.75 (91.50, 102.00), *p* < 0.0001, between the United States and Japan in 2021 (Table 1), but the number of confirmed COVID-19 cases per 1000 population per year (95% confidence intervals) in a 1% increment of vaccination rate was only −0.74 (−1.29, −0.20), *p* = 0.007, for the United States and −1.48 (−1.95, −1.00), *p* < 0.0001, for Japan (Table 2).

Table 3 shows the association of the booster vaccination rate (i.e., the percentage of people who have received a booster dose) as of the end of March 2022 with the number of confirmed COVID-19 cases per 1000 population per year in 2022. In 2022, a higher booster vaccination rate was associated with a lower number of confirmed COVID-19 cases both in the United States and Japan before adjustment of the vaccination rate (i.e., the percentage of people who were fully vaccinated with the second dose of a two-dose vaccine or one dose of a single-dose vaccine), but the association was attenuated after adjustment, particularly for the United States. The number of confirmed COVID-19 cases per 1000 population per year (95% confidence intervals) in a 1% increment of the booster vaccination rate was −2.89 (−4.75, −1.04), *p* = 0.002, for the United States, and −7.94 (−12.06, −3.83), *p* = 0.0002, for Japan (*p* for interaction was 0.04) in the crude model (Figure 4), and the respective number was −0.40 (−4.18, 3.38), *p* = 0.83, for the United States, and −5.61(−9.86, −1.36), *p* = 0.01, for Japan (*p* for interaction was 0.08) in the multivariable model, whereas there was no variable with ≥5 VIF. In the multivariable model, the VIF was 4.32 for the United States and 1.21 for Japan, and the Pearson correlation coefficient between vaccination rate and booster vaccination rate was 0.88, *p* = 0.0001 for the United States, and 0.42, *p* = 0.003 for Japan. The adjusted R-square was 0.14 for the United States and 0.22 for Japan in the crude model, and the adjusted R-square was 0.16 and 0.30, respectively, in the multivariable model.

In the log-linear model, the associations were not altered substantially, although the interaction became large. The percent change in a 1% increment of the booster vaccination rate was −1.0% (−1.6%, −0.4%), *p* = 0.001, for the United States, and −6.5% (−9.2%, −3.7%), *p* < 0.0001 for the Japan (*p* for interaction was 0.0007) in the crude model.

## 4. Discussion

This study showed a macro-level association of the vaccination rate with the number of confirmed COVID-19 cases in both the United States and Japan. A higher vaccination rate was associated with a lower number of confirmed COVID-19 cases both in the United States and Japan in 2021, and the association was consistent. A higher booster vaccination rate was also associated with a lower number of confirmed COVID-19 cases both in the United States and Japan before adjustment for the vaccination rate (i.e., the percentage of people who were fully vaccinated with the second dose of a two-dose vaccine or one dose of a single-dose vaccine), although the association was attenuated particularly for the United States after adjustment. These findings suggest that vaccination may be effective in reducing SARS-CoV-2 infections in a macro-level analysis.

Although the messenger RNA (mRNA) vaccine has been shown to be effective in preventing symptomatic infections of SARS-CoV-2 in clinical trials under optimal conditions [35,36,37,38], its effectiveness in preventing SARS-CoV-2 infections (i.e., including asymptomatic infections) has been examined in real-world settings [5,6,7,8,9,10,11,12,13,14,15,16,17,18,19,20,21,22]. Observational studies in Israel showed that the first dose and the second dose of the BNT162b2 mRNA vaccine were effective for a wide range of COVID-19-related outcomes, including SARS-CoV-2 infections among residents [5,6] and among health care workers [7,8]. Observational studies in the United States showed that the first dose and the second dose of mRNA vaccine (i.e., BNT162b2 and mRNA-1273) were effective in preventing SARS-CoV-2 infections among health care personnel, first responders and other essential and frontline workers [9] and preventing asymptomatic SARS-CoV-2 infections among patients who underwent preprocedural and presurgical SARS-CoV-2 molecular testing [10] in early 2021. Another observational study in the United States also showed that the first dose and the second dose of the BNT162b2 mRNA vaccine were effective in preventing symptomatic and asymptomatic infections of SARS-CoV-2 among hospital employees [11]. Similarly, among health care workers, short-term protection by the first dose of the BNT162b2 mRNA vaccine against asymptomatic infections of SARS-CoV-2 in the United Kingdom [12] and the effectiveness of the first dose and second dose of the BNT162b2 mRNA vaccine in preventing SARS-CoV-2 infections in Italy [13] were observed in early 2021. In Spain, an observational study among adults who were close contacts of COVID-19 cases showed that the vaccine effectiveness against SARS-CoV-2 infections ranged from 50% to 86% between April and August 2021 [14]. These findings indicate the effectiveness of vaccination in preventing SARS-CoV-2 infections in individual-level analyses during an early period of the COVID-19 pandemic after the introduction of vaccination in real-world settings.

Other observational studies in the middle of 2021 showed that the effectiveness of the second dose of the mRNA vaccine against SARS-CoV-2 infections might wane after a few months, although it remained at modest levels [15,16,17]. An observational study of adults fully vaccinated in Israel showed that protection against infections of the Delta variant waned in all ages a few months after receiving the second dose of the BNT162b2 mRNA vaccine [15]. An observational study of residents in Qatar showed that the second dose of the BNT162b2 mRNA vaccine was effective in preventing infections of the Beta and Delta variants, but it appeared to wane rapidly following its peak after the second dose [16]. An observational study in members of health care services in the United States showed that the effectiveness of the BNT162b2 mRNA vaccine in preventing infections of Delta variant was 93% during the first month after the second dose vaccination but declined to 53% after 4 months, although the effectiveness against non-Delta variants also declined to 67% at 4-5 months [17]. Following these findings, previous studies showed that a booster dose of the BNT162b2 mRNA vaccine was associated with a lower rate of SARS-CoV-2 infections among health care workers [18] and among residents aged ≥60 years old [19] in Israel. During this period, some studies started to indicate that vaccination may be imperfect as a countermeasure against the COVID-19 pandemic, but these studies also show the effectiveness of vaccination in reducing SARS-CoV-2 infections.

Even during an Omicron predominant period, previous studies showed the effectiveness of a booster dose following the two-dose initial series in preventing symptomatic and asymptomatic infections after December 2021 [20,21]. A test-negative case-control study in the United States showed high and durable effectiveness of the booster dose of the mRNA-1273 mRNA vaccine in preventing infections of Delta variant, but lower effectiveness against Omicron variant among members of health care services, although the effectiveness against Omicron variant was 71.6% at 14-60 days and 47.4% at >60 days after vaccination [20]. A retrospective cohort study of residents in Qatar showed that a booster dose of the mRNA vaccine (i.e., BNT162b2 and mRNA-1273) was effective in preventing symptomatic infections of the Delta variant but less effective against the Omicron variant, although the effectiveness against the Omicron variant was 49.4% for the BNT162b2 mRNA vaccine and 47.3% for the mRNA-1273 mRNA vaccine [21]. These consistent findings suggest that vaccination is effective at preventing SARS-CoV-2 infections in individual-level analyses even during an Omicron predominant period. This study extends the previous results in individual-level analyses to that in a macro-level analysis.

On the other hand, a simple international comparison showed that a higher vaccination rate was associated with a higher number of confirmed COVID-19 cases per 1000 population per year in the world, suggesting that this kind of simple international comparison is more likely to be affected by potential confounding factors. In fact, whereas a macro-level analysis showed consistent associations of higher vaccination rates with a lower number of confirmed COVID-19 cases for the United States and Japan based on jurisdiction data, the effect was smaller than that in the national difference. These findings suggest that it may be difficult to detect the effectiveness of vaccination in a simple international comparison using data from each country due to the effect of national differences.

In this study, the association of booster vaccination rate with the number of confirmed COVID-19 cases per year was attenuated after adjustment of the vaccination rate (i.e., the percentage of people who were fully vaccinated with the second dose of a two-dose vaccine or one dose of a single-dose vaccine), particularly for the United States. Although multicollinearity could not be observed (i.e., VIFs < 5), the booster vaccination rate was strongly associated with the vaccination rate, particularly for the United States, suggesting that it may be difficult to divide the effect of the booster vaccination rate from that of the vaccination rate in the multivariable model for the United States. On the other hand, as the booster dose is received against wane of the effect in the second dose of a two-dose vaccine or one dose of a single-dose vaccine, the booster vaccination rate is more likely to contribute to reducing SARS-CoV-2 infections rather than the vaccination rate after the start of the booster vaccination. These findings suggest that it may be more suitable to show the association of booster vaccination rate with the number of confirmed COVID-19 cases in a crude model.

One of the potential mechanisms regarding the association of a higher vaccination rate with a lower number of confirmed COVID-19 cases is the effect of vaccination in preventing SARS-CoV-2 infections among persons who were vaccinated, as mentioned above. Another individual-level potential mechanism is the effect of vaccination on reducing transmission of SARS-CoV-2 infections. As with all infectious diseases, the pathogen is required to cause the disease, and the viral load in individuals who are vaccinated and have a breakthrough SARS-CoV-2 infection is substantially lower than that in unvaccinated people who developed infection [39]. Previous studies also showed that the viral lord declined more rapidly in vaccinated people than in unvaccinated people [40,41]. As viral load can be a proxy for infectiousness, vaccination may be effective in reducing the transmission of SARS-CoV-2 infections [42]. Moreover, from the viewpoint of multilevel analysis, the effects on individual-level outcomes can be divided into individual-level and group-level factors [43,44]. As SARS-CoV-2 infection is the sine qua non for disease transmission, the lower the number of people with SARS-CoV-2 infection is, the lower the chance of transmission becomes. It is difficult to achieve herd immunity [45], and details of transmission may be complex due to overdispersion [46] and heterogeneity [47] of SARS-CoV-2 infections, but a higher vaccination rate may also lead to reducing the chance of infections from the group-level effect beyond the individual-level effect in reducing infections and transmissions. These findings suggest that vaccination may have a group-level effect as well as an individual-level effect in reducing SARS-CoV-2 infections.

The findings of this study suggest that the vaccination rate is important for explanations of regional disparity of the COVID-19 pandemic in each country, which provides scientific evidence supporting the use of vaccination as a countermeasure against the COVID-19 pandemic, but limitations must be discussed. First, this study is an observational study using aggregate data, which may lead to potential confounding. For example, potential confounding factors such as temperature, the accuracy of registry and countermeasures against COVID-19 may affect the association. This study showed consistency in the association of vaccination rate with the number of confirmed COVID-19 cases per year using data from each jurisdiction for the United States and Japan, but the generalizability of the association is not examined in the other countries. Although a lot of previous studies in individual-level analyses support the effectiveness of vaccination in preventing SARS-CoV-2 infections [5,6,7,8,9,10,11,12,13,14,15,16,17,18,19,20,21,22], further studies are needed to confirm the generalizability of the macro-level association of the effectiveness of vaccination in reducing SARS-CoV-2 infections in other countries.

Second, the effects of the waning of vaccine protection and the variation of variants were not examined in this study. The design is a cross-sectional study but not a longitudinal study using a time-series dataset, and lag time is not included in this analysis. A previous meta-analysis indicates that the vaccine effectiveness against SARS-CoV-2 infections decreased from 1 to 6 months after full vaccination by 21.0 percent points [22], and previous studies also showed that booster vaccination was less effective against the Omicron variant than that against the Delta variant [20,21]. However, the progress of vaccination and the spread of variants are similar on a yearly basis among jurisdictions in each country, which may make these effects small in this macro-level analysis. Further detailed studies are needed using a time-series dataset to confirm the detailed effect of vaccination rate in reducing SARS-CoV-2 infections.

## 5. Conclusions

This study provided macro-level evidence that vaccination may reduce the number of confirmed COVID-19 cases. The magnitude may be smaller than that in the national difference, but the association was consistent in the United States and Japan, which provides scientific evidence supporting the use of vaccination as a countermeasure against the COVID-19 pandemic. These findings suggest that vaccination may be imperfect but effective in reducing SARS-CoV-2 infections.

## Figures and Tables

**Figure 1 ijerph-19-07435-f001:**
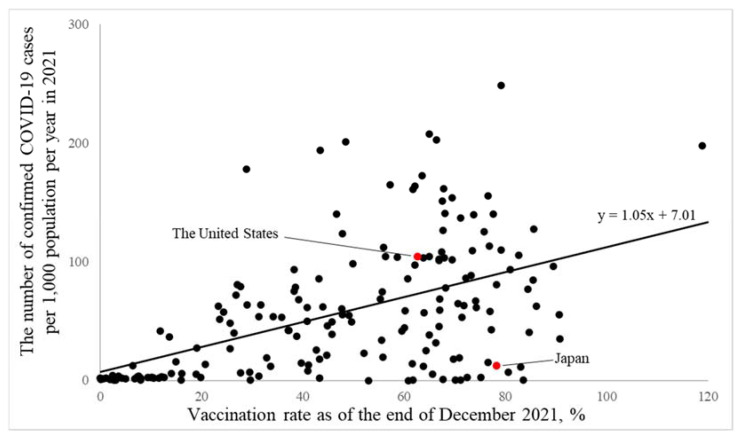
An international association of vaccination rate with the number of confirmed COVID-19 cases per year in 2021 in the world. The line shows the slope based on the generalized regression model as an approximated curve. The covariate was not included in this model.

**Figure 2 ijerph-19-07435-f002:**
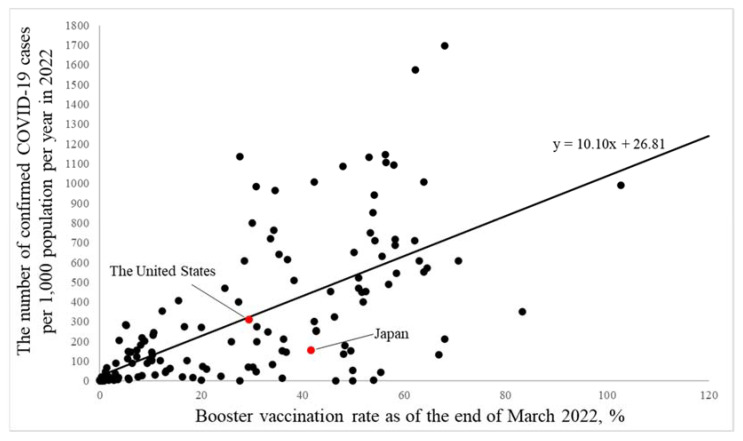
An international association of booster vaccination rate with the number of confirmed COVID-19 cases per year in 2022 in the world. The line shows the slope based on the generalized regression model as an approximated curve. The covariate was not included in this model.

**Figure 3 ijerph-19-07435-f003:**
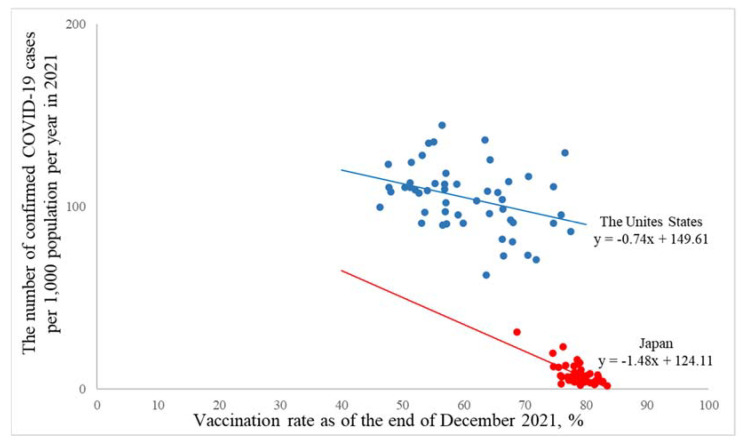
A macro-level association of the vaccination rate with the number of confirmed COVID-19 cases per year based on the jurisdictions in the United States and Japan. Blue dots are jurisdictions in the United States, and red dots are jurisdictions in Japan. The line shows the slope based on the generalized regression model as an approximated curve. The covariate was not included in this model.

**Figure 4 ijerph-19-07435-f004:**
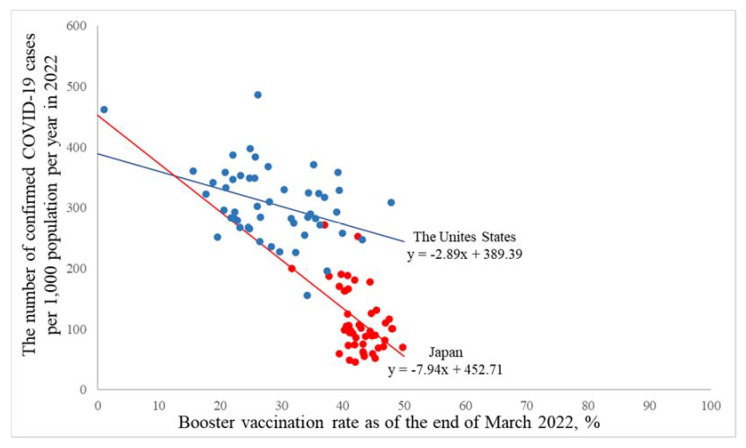
A macro-level association of booster vaccination rate with the number of confirmed COVID-19 cases per year based on the jurisdictions in the United States and Japan. Blue dots are jurisdictions in the United States, and red dots are jurisdictions in Japan. The line shows the slope based on the generalized regression model as an approximated curve. The covariate was not included in this model.

**Table 1 ijerph-19-07435-t001:** The number of vaccinated people and confirmed COVID-19 cases based on the jurisdictions in the United States and Japan.

	No.*	Mean (95% Confidence Intervals)		
		In 2020	In 2021	In 2022 **
**The number of people who were fully vaccinated, million persons**				
The United States	51	0.00(0.00, 0.00)	3.95(2.65, 5.25) ♰	0.22(0.13, 0.32) ♰
Japan	47	0.00(0.00, 0.00)	2.09(1.48, 2.70) ♰	0.05(0.03, 0.07) ♰
Difference		0.00(0.00, 0.00)	1.86(0.40, 3.32) §	0.17(0.07, 0.27) §
**Vaccination rate as of the last day, %**				
The United States	51	0.00(0.00, 0.00)	60.49(58.18, 62.80) ♰	62.64(59.30, 65.98) ♰
Japan	47	0.00(0.00, 0.00)	78.60(77.87, 79.34) ♰	80.18(79.47, 80.90) ♰
Difference		0.00(0.00, 0.00)	−18.11(−20.59, −15.64) §	−17.54(−21.04, −14.04) §
**The number of people who have received a booster dose, million persons**				
The United States	51	0.00(0.00, 0.00)	1.35(0.92, 1.78) ♰	0.52(0.30, 0.74) ♰
Japan	47	0.00(0.00, 0.00)	0.02(0.02, 0.03) ♰	1.10(0.78, 1.41) ♰
Difference		0.00(0.00, 0.00)	1.33(0.89, 1.77) §	−0.57(−0.96, −0.18) §
**Booster vaccination rate as of the last day, %**				
The United States	51	0.00(0.00, 0.00)	21.25(19.59, 22.91) ♰	28.39(26.12, 30.65) ♰
Japan	47	0.00(0.00, 0.00)	0.92(0.80, 1.04) ♰	42.71(41.78, 43.64) ♰
Difference		0.00(0.00, 0.00)	20.33(18.63, 22.04) §	−14.32(−16.82, −11.83) §
**The number of confirmed COVID-19 cases per year, million persons**				
The United States	51	0.38(0.27, 0.50)	0.66(0.46, 0.85) ♰	1.91(1.26, 2.57) ♰
Japan	47	0.00(0.00, 0.01)	0.03(0.02, 0.05) ♰	0.42(0.22, 0.61) ♰
Difference		0.38(0.26, 0.50) §	0.62(0.42, 0.82) §	1.50(0.80, 2.19) §
**The number of confirmed COVID-19 cases per 1000 population per year**				
The United States	51	63.11(56.99, 69.22)	104.65(99.77, 109.53) ♰	307.24(290.57, 323.91) ♰
Japan	47	1.08(0.81, 1.35)	7.90(6.26, 9.54) ♰	113.39(98.05, 128.73) ♰
Difference		62.03(55.76, 68.29) §	96.75(91.50, 102.00) §	193.80(170.80, 216.90) §

*: The number of jurisdictions; ♰: *p* for difference from 2020 was < 0.05; §: *p* for difference between the United States and Japan was < 0.05; **: Data as of 31 March 2022 was used.

**Table 2 ijerph-19-07435-t002:** The association of the vaccination rate with the number of confirmed COVID-19 cases per year for the United States and Japan.

	The Number of Confirmed Cases per 1000 Population per Year (95% Confidence Intervals) in 2021	
	The United States	Japan
**No. of jurisdictions**	51	47
**The number in 1% increment of vaccination rate ***		
Crude model **	−0.74(−1.29, −0.20) ♰	−1.48(−1.95, −1.00) ♰

*: The percentage of people who were fully vaccinated with the second dose of a two-dose vaccine or one dose of a single-dose vaccine as of the end of December 2021; ♰: *p* < 0.01; **: The covariate was not included in the crude model.

**Table 3 ijerph-19-07435-t003:** The association of booster vaccination rate with the number of confirmed COVID-19 cases per year for the United States and Japan.

	The Number of Confirmed Cases per 1000 Population per Year (95% Confidence Intervals) in 2022	
	The United States	Japan
**No. of jurisdictions**	51	47
**The number in 1% increment of booster vaccination rate**		
Crude model **	−2.89(−4.75, −1.04) ♰	−7.94(−12.06, −3.83) ♰
Multivariable model ***	−0.40(−4.18, 3.38)	−5.61(−9.86, −1.36) ♰
**The number in 1% increment of vaccination rate ***		
Crude model **	−2.17(−3.40, −0.93) ♰	−10.31(−15.68, −4.94) ♰
Multivariable model ***	−1.93(−4.49, 0.63)	−7.25(−12.78, −1.71) ‡

*: The percentage of people who were fully vaccinated with the second dose of a two-dose vaccine or one dose of a single-dose vaccine as of the end of March 2022; ♰: *p* < 0.01 ‡: *p* < 0.05; **: The covariate was not included in the crude model; ***: Both of vaccination rate and booster vaccination rate were included in the model as each covariate.

## Data Availability

Publicly available datasets were analyzed in this study. Data on the total number of people who were fully vaccinated in each jurisdiction where a recipient lives can be found at https://www.cdc.gov/coronavirus/2019-ncov/vaccines/distributing/about-vaccine-data.html (accessed on 3 April 2022) for the United States and at https://www.kantei.go.jp/jp/headline/kansensho/vaccine.html (accessed on 4 January 2022 and 1 April 2022) for Japan. Data on the total number of people who have received a booster dose can be found at https://www.cdc.gov/coronavirus/2019-ncov/vaccines/distributing/about-vaccine-data.html (accessed on 3 April 2022) for the United States and at https://info.vrs.digital.go.jp/dashboard (accessed on 9 April 2022) for Japan. Data on the cumulative number of confirmed COVID-19 cases can be found at https://data.cdc.gov/Case-Surveillance/United-States-COVID-19-Cases-and-Deaths-by-State-o/9mfq-cb36 (accessed on 3 April 2022) for the United States and at https://covid19.mhlw.go.jp/ (accessed on 3 April 2022) for Japan. Data on the total number of people who were fully vaccinated, data on the total number of people who have received a booster dose and data on the cumulative number of confirmed COVID-19 cases in each country and area can be found at https://ourworldindata.org/coronavirus (accessed on 5 April 2022).

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
