# Peer review of "A Macro-Level Association of Vaccination Rate with the Number of Confirmed COVID-19 Cases in the United States and Japan"

_ijerph, 2022, doi:10.3390/ijerph19127435_

Round 1
Reviewer 1 Report
The author attempted to provide evidence on the association of vaccination rates against reduction of SARS-CoV-2 infection rates. While the topic is important, i have several serious concerns which do not allow me to suggest accept of this manuscript:
Novelty: it has been mentioned already, but it is known that such a relationship exists. The interesting aspect here would have been to incorporate many more external factors, including environmental, sociodemographic, economic etc so as to accurately quantify the impact of vaccination. I do not see the added value of the obtained result here.
Data collection, usage and completeness: in such a multi-factorial phenomenon, with many additional factors known to play a dramatic role (lockdown levels, weather parameters like temperature and humidity, pollen, weekend and other registry errors, latitude etc), i cannot perceive that such an over-simplification may accurately yield a relationship. I agree with the existence of the relationship, but quantifying it via a GZM and only a couple of datasets is not convincing.
Robustness of statistical analyses: with an oversimplification of data usage of course there is a simplification of data analysis. More profound analyses, including also more data, would provide more reliable results. For example, when cumulative values of infection (and vaccination) rates are used, you are missing smaller-scale effects, like lockdowns, environmental effects etc, as already known by the multi-modal development of COVID curves per year. What about lag effects, what about artefacts (i.e. sensitivity analysis?), collinearity (mentioned but not shown, with other, non-included factors) and so on?
Author Response
FOR REVIEWER1
The author attempted to provide evidence on the association of vaccination rates against reduction of SARS-CoV-2 infection rates. While the topic is important, i have several serious concerns which do not allow me to suggest accept of this manuscript:
>> Thank you very much for valuable comments. I revised my manuscript point by point as follows.
Novelty: it has been mentioned already, but it is known that such a relationship exists. The interesting aspect here would have been to incorporate many more external factors, including environmental, sociodemographic, economic etc so as to accurately quantify the impact of vaccination. I do not see the added value of the obtained result here.
>> Thank you very much for valuable comment. As reviewer mentioned, this theme has been mentioned already in daily discussion. But, in search of PubMed, there is no article which shows the associations which this article mentioned as comparison in the United States and Japan. To my knowledge (as well as a search of PubMed), this is the first report of a macro-level association of vaccination rate with SARS-CoV-2 infections as comparison in the United States and Japan as article. I agree with reviewer’s comment about importance of external factors. Therefore, I added the limitation as follows; “First, design of this study is an observational study using aggregate data, which may lead to potential confounding. For example, potential confounding factors such as temperature, accuracy of registry and countermeasures against COVID-19 may affect the association.” (P11L378).
Data collection, usage and completeness: in such a multi-factorial phenomenon, with many additional factors known to play a dramatic role (lockdown levels, weather parameters like temperature and humidity, pollen, weekend and other registry errors, latitude etc), i cannot perceive that such an over-simplification may accurately yield a relationship. I agree with the existence of the relationship, but quantifying it via a GZM and only a couple of datasets is not convincing.
>> Thank you very much for valuable comment. As reviewer mentioned, I agree with importance of additional factors. Therefore, I added the limitation as follows; “First, this study is an observational study using aggregate data, which may lead to potential confounding. For example, potential confounding factors such as temperature, accuracy of registry and countermeasures against COVID-19 may affect the association.” (P11L378). I also agree with the limitation of usage of only a couple of datasets, I therefore added the limitation as follows; ” This study showed consistency in the association of vaccination rate with the number of confirmed COVID-19 cases per year using data in each jurisdiction for the United States and Japan, but generalizability of the association is not examined in the other countries. Although a lot of previous studies in individual-level analyses support effectiveness of vaccination in preventing SARS-CoC-2 infections [5-22], further studies were needed to confirm generalizability of the macro-level association in effectiveness of vaccination in reducing SARS-CoV-2 infections in other countries.” (P11L381), “The design is a cross-sectional study, but not a longitudinal study using time series dataset, and lag time is not included in this analysis.” (P11L389) and “Further detailed studies were needed using time series dataset to confirm detailed effect of vaccination rate in reducing SARS-CoV-2 infections.” (P11L396)
Robustness of statistical analyses: with an oversimplification of data usage of course there is a simplification of data analysis. More profound analyses, including also more data, would provide more reliable results. For example, when cumulative values of infection (and vaccination) rates are used, you are missing smaller-scale effects, like lockdowns, environmental effects etc, as already known by the multi-modal development of COVID curves per year. What about lag effects, what about artefacts (i.e. sensitivity analysis?), collinearity (mentioned but not shown, with other, non-included factors) and so on?
>> Thank you very much for valuable comment. As reviewer mentioned, I agree with importance of further profound studies using to confirm smaller-scale effects including lag times. Therefore, I added the descriptions as follows; “The design is a cross-sectional study, but not a longitudinal study using time series dataset, and lag time is not included in this analysis. A previous meta-analysis indicates that vaccine effectiveness against SARS-CoV-2 infections decreased from 1 month to 6 months after full vaccination by 21.0 percent points [22], and previous studies also showed that booster vaccination was less effective against Omicron variant than that against Delta variant [20,21]. However, progress of vaccination and spread of variants are similar on a yearly basis among jurisdictions in each country, which may make these effects small in this macro-level analysis. Further detailed studies were needed using time series dataset to confirm detailed effect of vaccination rate in reducing SARS-CoV-2 infections.” (P11L389) for further profound analysis and lag times, “In multivariable model, the VIF was 4.32 for the United States and 1.21 for Japan, and Pearson correlation coefficient between vaccination rate and booster vaccination rate was 0.88, p=0.0001 for the United States and 0.42, p=0.003 for Japan.” (P8L241) and “multicollinearity could not be observed (i.e., VIFs<5)” (P10L346) for collinearity, “First, this study is an observational study using aggregate data, which may lead to potential confounding. For example, potential confounding factors such as temperature, accuracy of registry and countermeasures against COVID-19 may affect the association.” (P11L378) for non-included factors.
Reviewer 2 Report
Overall, I found this paper a fascinating study of the data on COVID-19 cases and vaccination, showing that the latter had a measurable impact on the number of cases.
Intro
Lines 36-38: I think the reasons for concerns of waning immunity and the threat posed by the variants could be expanded upon. These are arguably two of the biggest risks (along with vaccine hesitancy) to vaccine efficacy on a global scale.
The phrase macro-level is used frequently but I think it could be described more on first usage at line 41.
Lines 48-52: While I agree with this statement of the concerns of vaccine hesitancy, I think it needs to be added that there is a mixed message about what is meant by vaccine efficacy. The widespread idea that vaccines have not stopped people getting infected shows a lack of understanding of what they do; as I frequently explain, vaccines do not give you an invisible shield to prevent infection but make it less likely that you will get sick, and this may also impact transmission and case numbers.
Materials and Methods
Publicly available data sets were used and references to these are given.
While I think the international comparison is good, I wonder about how it fits with the remit of the study. Perhaps this could be better integrated into the aim of the paper?
For the vaccination rate, could you specify the age group(s) covered? Since this changed in some countries at least part way through the period under study, I think this should be stated.
Results
The results show the inverse relationship between vaccination and COVID-19 cases in the US and Japan, as well as an association between booster shots and reduced COVID-19 cases.
Figures
Good – but does the vaccination % refer to e.g. all adults or anyone over 16 or 12 years of age?
Discussion
I think the analysis and interpretation are effective at situating the data in the wider context. I do wonder about the merit of the international study, not least given the authors’ own comments on the limitations of this part of the paper (Lines 294-296).
Writing and references:
The writing standard is generally good though it would help to have a native speaker editor proofread it. I highlighted some of the issues I spotted on the PDF of the manuscript.
The reference list includes important developments (e.g. the SARS-Cov-2 vaccine trials), public data resources and otherwise relevant literature.
Author Response
FOR REVIEWER2
Overall, I found this paper a fascinating study of the data on COVID-19 cases and vaccination, showing that the latter had a measurable impact on the number of cases.
>> Thank you very much for valuable comments. I revised my manuscript point by point as follows.
Intro
Lines 36-38: I think the reasons for concerns of waning immunity and the threat posed by the variants could be expanded upon. These are arguably two of the biggest risks (along with vaccine hesitancy) to vaccine efficacy on a global scale.
>>I added the description as follows; “In fact, a previous study shows that the effectiveness against Omicron variant was 71.6% at 14-60 days and 47.4% at >60 days after vaccination [20], and another study shows that booster dose of mRNA vaccine (i.e., BNT162b2 and mRNA-1273) was effective in pre-venting symptomatic infections of Delta variant, but less effective against Omicron variant, although the effectiveness against Omicron variant was 49.4% for BNT162b2 mRNA vaccine and 47.3% for mRNA-1273 mRNA vaccine [21].” (P1L38).
The phrase macro-level is used frequently but I think it could be described more on first usage at line 41.
>>I added the description as follows; “an analysis using population, but not individual, as each observation in dataset” (P1L46).
Lines 48-52: While I agree with this statement of the concerns of vaccine hesitancy, I think it needs to be added that there is a mixed message about what is meant by vaccine efficacy. The widespread idea that vaccines have not stopped people getting infected shows a lack of understanding of what they do; as I frequently explain, vaccines do not give you an invisible shield to prevent infection but make it less likely that you will get sick, and this may also impact transmission and case numbers.
>>I added the description as follows; “Moreover, mixed message about effectiveness of vaccine (e.g., the difference of effectiveness between SARS-CoV-2 infections and COVID-19 severe diseases) may be affect the vaccine acceptance, because the effectiveness of vaccine in reducing SARS-CoV-2 infections may be weaker than that in reducing COVID-19 severe diseases [22].” (P2L47).
Materials and Methods
Publicly available data sets were used and references to these are given.
>> Thank you very much for valuable comments. As reviewer mentioned, the references to the public available data sets are cited as reference number 27-32. (P13L496)
While I think the international comparison is good, I wonder about how it fits with the remit of the study. Perhaps this could be better integrated into the aim of the paper?
>>I added the purpose of international comparison as follows; “Aiming to show global position of the United States and Japan in addition to difference between intranational and international associations of vaccination rate with the number of confirmed COVID-19 cases, a simple international comparison was also conducted to examine an association of vaccination rate and booster vaccination rate with the number of confirmed COVID-19 cases in the world.” (P3L99).
For the vaccination rate, could you specify the age group(s) covered? Since this changed in some countries at least part way through the period under study, I think this should be stated.
>> I added the definition of vaccination rate as follows; ”In calculation of vaccination rate, the number of overall population was used as denominator.” (P4L124). Although persons who could not receive vaccination (e.g., children) may be included as denominator, I included these persons as denominator of vaccination rate, as these persons were not protected by vaccination.
Results
The results show the inverse relationship between vaccination and COVID-19 cases in the US and Japan, as well as an association between booster shots and reduced COVID-19 cases.
>> Thank you very much for valuable comments. As reviewer mentioned, I showed the inverse associations.
Figures
Good – but does the vaccination % refer to e.g. all adults or anyone over 16 or 12 years of age?
>> I added the definition of vaccination rate as follows; ”In calculation of vaccination rate, the number of overall population was used as de-nominator.” (P4L124). Although persons who could not receive vaccination (e.g., children) may be included as denominator, I included these persons as denominator of vaccination rate, as these persons were not protected by vaccination.
Discussion
I think the analysis and interpretation are effective at situating the data in the wider context. I do wonder about the merit of the international study, not least given the authors’ own comments on the limitations of this part of the paper (Lines 294-296).
>> Thank you very much for valuable comments. I added the purpose of international comparison as follows; “Aiming to show global position of the United States and Japan in addition to difference between intranational and international associations of vaccination rate with the number of confirmed COVID-19 cases, a simple international comparison was also conducted to examine an association of vaccination rate and booster vaccination rate with the number of confirmed COVID-19 cases in the world.” (P3L99) as well as the paragraph in discussion section as follows; “On the other hand, a simple international comparison showed that higher vaccination rate was associated with higher number of confirmed COVID-19 cases per 1,000 population per year in the world, suggesting that this kind of simple international comparison is more likely to be affected by potential confounding factors. In fact, whereas a macro-level analysis showed consistent associations of higher vaccination rate with lower number of confirmed COVID-19 cases for the United States and Japan based on jurisdiction data, the effect was smaller than that in the national difference. These findings suggest that it may be difficult to detect effectiveness of vaccination in a simple international comparison using data in each country due to effect of national differences.” (P10L333).
Writing and references:
The writing standard is generally good though it would help to have a native speaker editor proofread it. I highlighted some of the issues I spotted on the PDF of the manuscript.
>>The manuscript was edited again, and I changed the description as follows; “data of cumulative number of confirmed COVID-19 cases were obtained from Web site of national governments.” (P1L13). “This study provided macro-level evidence that vaccination may reduce the number of confirmed COVID-19 cases.” (P1L21), “but many vaccines confer only short-term and modest levels of individual protection [2-4].” (P1L28), “Therefore, vaccine is not always ultimate solution, and vaccination also has been implemented as one of the countermeasures against some of infectious diseases” (P1L29), “However, there is limited evidence to the masses which showed that vaccination reduced the number of confirmed COVID-19 cases in a macro-level analysis (i.e., an analysis using population, but not individual, as each observation in dataset),” (P1L44), “but more likely to be affected by potential confounding factors such as population size and ideology in each country.” (P2L55), “Data of total number of people who were fully vaccinated as of the end of March 2022 was also used to adjust as potential confounding factor.” (P2L93), “suggesting that this kind of simple international comparison is more likely to be affected by potential confounding factors.” (P10L335), “These findings suggest that vaccination may have group-level effect as well as individual-level effect in reducing SARS-CoV-2 infections.” (P10L372), “but limitations must be discussed.” (P11L378), “Although a lot of previous studies in individual-level analyses support effectiveness of vaccination in preventing SARS-CoV-2 infections [5-22],” (PL384), and “Data of total number of people who have received a booster dose can be found” (P11L413). I deleted the sentences “Vaccines are imperfect, but probably effective in reducing SARS-CoV-2 infections.” and “but previous studies suggest that disbelief in efficacy and effectiveness of vaccine can be one of the root causes of vaccine hesitancy [29,30].” for the reviewer’s comment. For the sentence “These consistent findings suggest that vaccination have effectiveness in preventing SARS-CoV-2 infections in individual-level analyses even during an Omicron predominant period.” (P10L329), I would like to use the word “vaccination”, because this sentence means that “vaccination” as countermeasure but not “vaccine” as product, may have effectiveness in preventing SARS-CoV-2 infections in individual-level analyses even during an Omicron predominant period. For the sentence “First, this study is an observational study using aggregate data, which may lead to potential confounding.” (P11L379), I would like to use the word “confounding”, but not “confounding factor”, because this sentence means that the study design may lead to “potential confounding” as phenomenon, but not “potential confounding factor” as variable.
The reference list includes important developments (e.g. the SARS-Cov-2 vaccine trials), public data resources and otherwise relevant literature.
>> Thank you very much for valuable comments.
Reviewer 3 Report
In this work, the authors perform a cross-sectional study on Data from the United States of America and Japan to evaluate a macro-level association of vaccination rate as well as booster vaccination rate with the number of confirmed COVID-19 cases. They use the generalized regression model to examine the association. The results of their analysis show that a higher vaccination rate is associated with a lower number of confirmed COVID-19 cases per year in 2021 for both the United States and Japan.
The manuscript is well presented and the results seem correct. Nevertheless, one point must be clarified:
Why do the authors use linear regression? What happens if the analysis is conducted with non-linear regression?
Author Response
FOR REVIEWER3
In this work, the authors perform a cross-sectional study on Data from the United States of America and Japan to evaluate a macro-level association of vaccination rate as well as booster vaccination rate with the number of confirmed COVID-19 cases. They use the generalized regression model to examine the association. The results of their analysis show that a higher vaccination rate is associated with a lower number of confirmed COVID-19 cases per year in 2021 for both the United States and Japan.
>> Thank you very much for valuable comments. I revised my manuscript point by point as follows.
The manuscript is well presented and the results seem correct. Nevertheless, one point must be clarified:
Why do the authors use linear regression? What happens if the analysis is conducted with non-linear regression?
>>I added the result of log-linear model (i.e., non-linear regression model) as sensitivity analysis as follows; “In log-linear model, the associations were not altered substantially, although the interaction become large. The percent change in 1% increment of vaccination rate was -0.7%(-1.2%, -0.2%), p=0.008, for the United States and -13.7%(-16.3%, -11.1%), p<0.0001 for the Japan (p for interaction was 0.003).” (P5L186) and “In log-linear model, the associations were not altered substantially, although the interaction become large. The percent change in 1% increment of booster vaccination rate was -1.0% (-1.6%, -0.4%), p=0.001, for the United States and -6.5% (-9.2%, -3.7%), p<0.0001 for the Japan (p for interaction was 0.0007) in crude model.” (P8L247) in addition to the description of method “As sensitivity analysis, log-linear model was also used in this analysis.” (P4L127).
Reviewer 4 Report
Thank you for the opportunity to revise this interesting paper titled: "A macro-level association of vaccination rate with the number of confirmed COVID-19 cases in United States and Japan".
It is an original paper with an interesting data analysis that compares the vaccination rate and infection rate of Japan and the USA.
The manuscript is valuable, but I would like to suggest some minor revisions. Please see below.
Title: United States needs "the" before. Please add.
The introduction can be shortened. Vaccination hesitancy is not the topic of this paper, so it can be reduced, as well as please consider that literature on the topic is now concordat in avoiding the term hesitancy, and instead, using the term vaccine acceptance. Revise the introduction and the manuscript accordingly.
The methods are well described. Please, add a section where describing the covariates used. Moreover, please specify the variable used in the adjusted model. Add it in the text and in the footnote of tables and figures.
Please, consider adding a graphical abstract in order to increase interest and readibility.
Discussion: some section of the discussion seem a mere list of previous evidence. Please, try to harmonize them in order to make them more discursive.
Limitations: please specify that you used aggregate data, and comment on their limits.
Conclusions: I think this work has an important public health impact, which results are very relevant in terms of public health policies and strategies. Add considerations on it.
Author Response
FOR REVIEWER4
Thank you for the opportunity to revise this interesting paper titled: "A macro-level association of vaccination rate with the number of confirmed COVID-19 cases in United States and Japan".
It is an original paper with an interesting data analysis that compares the vaccination rate and infection rate of Japan and the USA.
The manuscript is valuable, but I would like to suggest some minor revisions. Please see below.
>> Thank you very much for valuable comments. I revised my manuscript point by point as follows.
Title: United States needs "the" before. Please add.
>>I added “the” before “United States”.
The introduction can be shortened. Vaccination hesitancy is not the topic of this paper, so it can be reduced, as well as please consider that literature on the topic is now concordat in avoiding the term hesitancy, and instead, using the term vaccine acceptance. Revise the introduction and the manuscript accordingly.
>>I reduced the description of “vaccine hesitancy” and changed “vaccine hesitancy” into “vaccine acceptance” in the manuscript.
The methods are well described. Please, add a section where describing the covariates used. Moreover, please specify the variable used in the adjusted model. Add it in the text and in the footnote of tables and figures.
>>I added the description as follows; The covariate was not included in crude model, and both of vaccination rate and booster vaccination rate were included in multivariable model as each covariate.” (P3L125), “The covariate was not included in this model.” (P4L152), “The covariate was not included in this model.” (P5L167), “The covariate was not included in crude model.” (P7L213), “The covariate was not included in this model.” (P7L227), “The covariate was not included in crude model.” (P8L255), “Both of vaccination rate and booster vaccination rate were included in the model as each covariate.” (P8L256), and “The covariate was not included in this model.” (P9L298).
Please, consider adding a graphical abstract in order to increase interest and readibility.
>> I added a graphical abstract. (P2L57)
Discussion: some section of the discussion seem a mere list of previous evidence. Please, try to harmonize them in order to make them more discursive.
>>I added the description as follows; “These findings indicate effectiveness of vaccination in preventing SARS-CoV-2 infections in individual-level analyses during early period of the COVID-19 pandemic after introduction of vaccination in real-world settings.” (P9L291) and “During this period, some of studies started to indicate that vaccination may be imperfect as countermeasures against the COVID-19 pandemic, but these studies also show effectiveness of vaccination in reducing SARS-CoV-2 infections.” (P10L314).
Limitations: please specify that you used aggregate data, and comment on their limits.
>>I added limitation as follows; “First, this study is an observational study using aggregate data, which may lead to potential confounding. For example, potential confounding factors such as temperature, accuracy of registry and countermeasures against COVID-19 may affect the association.” (P11L378).
Conclusions: I think this work has an important public health impact, which results are very relevant in terms of public health policies and strategies. Add considerations on it.
>>I added the description as follows; “Findings of this study suggest that vaccination rate is important for explanation in regional disparity of the COVID-19 pandemic in each country, which provides scientific evidence supporting use of vaccination as countermeasure against the COVID-19 pandemic.” (P11L375) and “The magnitude may be smaller than that in the national difference, but the association was consistent in the United States and Japan, which provides scientific evidence supporting use of vaccination as countermeasure against the COVID-19 pandemic.” (P11L401)
Round 2
Reviewer 1 Report
The authors have addressed all my comments and the manuscript has been further improved. Given the limitations of the study and rather simplistic analysis, i am still concerned of the magnitude of the effect (which could be higher as a matter of fact), but i believe the topic is important and the message timely and has to be out to the public.
Reviewer 3 Report
This version of the manuscript is well improved. I recommend it for publication in the present form.